# Prompting and Evaluating Large Language Models for Proactive Dialogues: Clarification, Target-guided, and Non-collaboration

**Yang Deng**[1], **Lizi Liao**[2], **Liang Chen**[3], **Hongru Wang**[3], **Wenqiang Lei**[4], **Tat-Seng Chua**[1]

[1]National University of Singapore [2]Singapore Management University
[3]The Chinese University of Hong Kong [4]Sichuan University
{ydeng,dcscts}@nus.edu.sg, lzliao@smu.edu.sg
{lchen,hrwang}@se.cuhk.edu.hk, wenqianglei@gmail.com

## Abstract

Conversational systems based on Large Language Models (LLMs), such as ChatGPT, show exceptional proficiency in context understanding and response generation. However, they still possess limitations, such as failing to ask clarifying questions to ambiguous queries or refuse users' unreasonable requests, both of which are considered as key aspects of a conversational agent's proactivity. This raises the question of whether LLM-based conversational systems are equipped to handle proactive dialogue problems. In this work, we conduct a comprehensive analysis of LLM-based conversational systems, specifically focusing on three key aspects of proactive dialogues: clarification, target-guided, and non-collaborative dialogues. To trigger the proactivity of LLMs, we propose the Proactive Chain-of-Thought prompting scheme, which augments LLMs with the goal planning capability over descriptive reasoning chains. Empirical findings are discussed to promote future studies on LLM-based proactive dialogue systems.

## 1 Introduction

Conversational systems are envisioned to provide social support or functional service to human users via natural language interactions. Most research typically centers around a system's response capabilities, such as understanding the dialogue context (Wu et al., 2020; Chen et al., 2022; Deng et al., 2022b) and generating appropriate responses (Zhang et al., 2020b; Roller et al., 2021). The popularity of conversational systems has grown unprecedentedly with the advent of Chat-GPT, which showcases exceptional capabilities of context understanding and response generation with large language models (LLMs). Recent studies observe that, compared with current fine-tuned state-of-the-art (SOTA) methods, ChatGPT can still achieve competitive performance under zero-shot setting on different dialogue problems, such as the

knowledge-grounded dialogues (Bang et al., 2023), task-oriented dialogues (Zhang et al., 2023), and emotion-aware dialogues (Zhao et al., 2023).

Despite the strength of ChatGPT, there are still several limitations[1], such as failing to ask clarification questions to ambiguous user queries or refuse problematic user requests. These kinds of capabilities are typically regarded as the *proactivity* of the conversational system (Deng et al., 2023b), where the system can create or control the conversation to achieve the conversational goals by taking initiative and anticipating impacts on themselves or the human users. Thus, it raises the question: *Are these LLM-based conversational systems equipped to manage proactive dialogue problems?*

In this work, we conduct the first comprehensive analysis of LLM-based conversational systems on three common aspects of proactive dialogues, including 1) clarification in information-seeking dialogues (Guo et al., 2021; Deng et al., 2022a) where the system is required to proactively ask clarification questions when encountering ambiguity in user queries; 2) target-guided open-domain dialogues (Tang et al., 2019; Wu et al., 2019) where the system is required to proactively lead the conversation towards the designated target; and 3) non-collaborative task-oriented dialogues (Li et al., 2020; Zhou et al., 2020; Deng et al., 2023a) where the system and the user do not share the same conversational goal while the system aims to strategically reach a consensus with the user.

Motivated by the emergent capabilities of LLMs (Wei et al., 2022a,b) on reasoning over texts, some recent studies investigate in-context learning or chain-of-thought prompting schemes on planning (Huang et al., 2022) or taking actions (Yao et al., 2022) in interactive environments. Similarly, strategy learning and goal planning attach great importance in proactive dialogue systems. In order to enhance the proactivity of LLM-based conversa-

---

[1]as stated in https://openai.com/blog/chatgpt/.

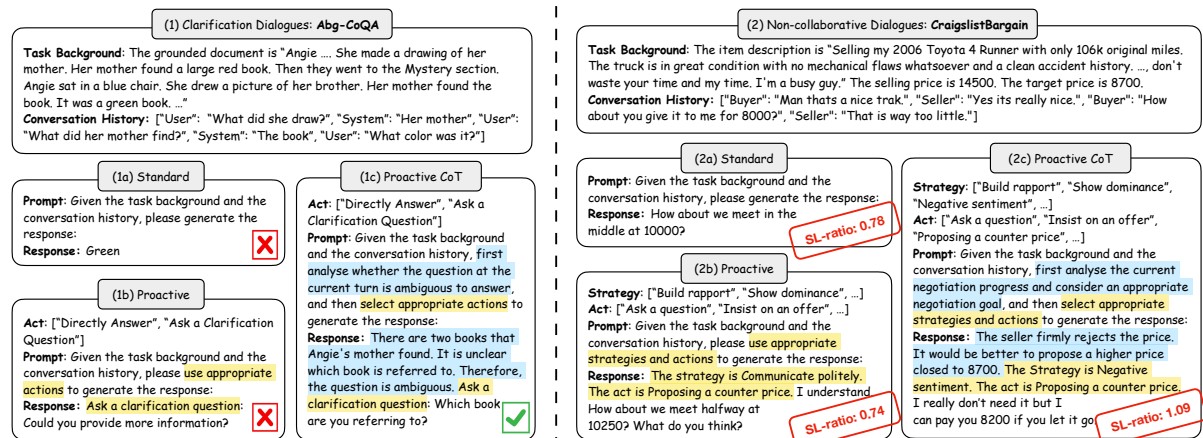

Figure 1: Examples of three kinds of prompting schemes for proactive dialogues. In the example of non-collaborative dialogue, the system plays the role of "Buyer", and the sale-to-list (SL) ratio shows the effectiveness of negotiation, which is calculated by (listed price − bargain price)/(listed price − buyer target price). The higher ratio means the current bargain price is closer to the target.

tional systems, we design the proactive chain-of-thought prompting (ProCoT) scheme. As shown in Figure 1, with standard prompting, LLM-based systems directly provide a randomly-guessed answer to the ambiguous user question (1a), or generate a general bargain response without any negotiation strategy (2a). When providing the system with options to take different dialogue acts (proactive prompting), the generated responses are unaware of the conversational goal, such as generating under-specified clarification questions (1b) and conservative negotiation responses (2b). To this end, ProCoT first instructs the system to generate descriptive thoughts about intermediate steps of reasoning and planning for reaching the conversational goal, and then make the decision of the next action to take. Finally, the system generates an appropriate response based on the decided action (1c & 2c).

We conduct extensive experiments with two LLM-based conversational systems, including ChatGPT and an open-sourced model, Vicuna (Chiang et al., 2023). With the aforementioned three types of prompting schemes, we compare these LLM-based conversational systems with fine-tuned SOTA dialogue models. The main contributions of this work can be summarized as follows:

- This work presents the first comprehensive evaluation on the proactivity of LLM-based dialogue systems, including the handling of clarification, target-guided, and non-collaborative dialogues.

- We design the proactive chain-of-thought prompting scheme to endow LLM-based dialogue systems with the capability of planning and taking

the initiative towards the conversational goal.

- Specifically, the main findings of the evaluation of LLM-based dialogue systems include: 1) They barely ask clarification questions when encountering ambiguous queries, and ProCoT largely overcomes this issue, though the performance is still unsatisfactory in domain-specific applications (§4.1). 2) They are proficient at performing topic shifting towards the designated target, but tend to make aggressive topic transition. ProCoT further improves this capability by planning a smoother transition (§4.2). 3) They fail to make strategic decision and tend to compromise with the opponent. The key challenge is how to effectively optimize the strategy learning (§4.3).

## 2 Related Works

**Proactive Dialogues.** Recent years have witnessed many advanced designs on developing proactive dialogue systems (Liao et al., 2023) for various applications. For example, target-guided dialogues aim to proactively lead the conversation to either a designated target topic (Tang et al., 2019) or a pre-defined knowledge entity (Wu et al., 2019). Existing studies typically adopt keyword transition (Qin et al., 2020; Zhong et al., 2021) or knowledge graph reasoning (Yang et al., 2022; Lei et al., 2022) techniques to proactively plan the topic thread towards the target. Besides, in information-seeking dialogues, proactive dialogue systems can ask clarification questions for clarifying the ambiguity of the query or question in conversational search (Aliannejadi et al., 2021) and

question answering (Guo et al., 2021; Deng et al., 2022a). In addition, under the non-collaborative setting, the system and the user have competing goals towards the task completion but the system aims to proactively reach an agreement favorable to itself (Zhou et al., 2020), such as negotiating a product price (He et al., 2018) or persuading users to make a donation (Wang et al., 2019).

**Large Language Models for Dialogues.** Previous dialogue systems, such as DialoGPT (Zhang et al., 2020b), Meena (Adiwardana et al., 2020), BlenderBot (Roller et al., 2021), LaMDA (Thoppilan et al., 2022), typically fine-tune pre-trained language models on public dialogue data. Inspired by the success of ChatGPT, recent practices build dialogue systems through conducting supervised fine-tuning on open-source large language models, such as LLaMA (Touvron et al., 2023), with either constructed instruction-following examples (*e.g.*, Alpaca (Taori et al., 2023)) or distilled conversation data (*e.g.*, Vicuna (Chiang et al., 2023)) from Chat-GPT. As all these LLM-based dialogue systems are trained to follow the user's instruction, it remains a question on whether these systems can take the initiative for handling proactive dialogues.

**Prompting in Dialogue Systems.** To induce knowledge from LLMs, various prompting methods are designed for zero-shot or few-shot learning in dialogue applications, such as task-oriented dialogues (Lee et al., 2021; Mi et al., 2022), knowledge-grounded dialogues (Shuster et al., 2022; Liu et al., 2022; Wang et al., 2023c), and open-domain dialogues (Chen et al., 2023b; Lee et al., 2023; Wang et al., 2023a). For example, Chen et al. (2023c) propose to prompt LLMs for controllable response generation in emotional support and persuasion dialogues, conditioned on the ground-truth dialogue strategies. In this work, we aim at prompting LLMs to proactively interact with the users.

## 3 Prompting LLMs to be Proactive

As presented in Figure 1, we describe the prompting schemes, including the standard, proactive, and proactive chain-of-thought (ProCoT) prompting.

**Standard Prompting.** In order to instruct LLMs to perform specific dialogue tasks, the typical prompting scheme can be formulated as

$$p(r|\mathcal{D}, \mathcal{C}). \quad (1)$$

Given the task background $\mathcal{D}$ and the conversation history $\mathcal{C}$, instruct the LLM to generate the response $r$. In specific, the task background can be the grounded document in clarification dialogues or the target description in target-guided dialogues.

**Proactive Prompting.** Proactive prompting aims to provide alternative options for LLMs to decide what kinds of actions should be taken in the response, instead of simply responding to the instruction. It can be formulated as:

$$p(a, r|\mathcal{D}, \mathcal{C}, \mathcal{A}). \quad (2)$$

Given the task background $\mathcal{D}$, the conversation history $\mathcal{C}$, and a set of possible dialogue acts $\mathcal{A}$, instruct the LLM to select the most appropriate dialogue act $a \in \mathcal{A}$ and then generate the response $r$. For example, the dialogue act can be *Ask a Clarification Question* or *Directly Answer the Question* in clarification dialogues, different negotiation strategies in non-collaborative dialogues, or different conversation topics in target-guided dialogues.

**Proactive Chain-of-Thought Prompting.** In order to endow LLMs with the capability of planning and taking the initiative towards the ultimate goal, we develop the proactive chain-of-thought prompting scheme—ProCoT. It involves the analysis of the next action to take by performing dynamic reasoning and planning for reaching the conversational goal. ProCoT can be formulated as:

$$p(t, a, r|\mathcal{D}, \mathcal{C}, \mathcal{A}), \quad (3)$$

where $t$ is the thought description for the decision-making process of the next action. For example, in clarification dialogues, $t$ can be the ambiguity analysis of the current user question as in Figure 1(1c). While in non-collaborative dialogues, $t$ can be the goal completion analysis of the current negotiation progress as in Figure 1(2c).

## 4 Evaluation

We evaluate the proactivity of LLM-based conversational systems from three perspectives, including the capability of asking clarification questions (§ 4.1), guiding the conversation towards the designated target (§ 4.2), and strategically handling conflicting goals (§ 4.3).

### 4.1 Clarification Dialogues

Clarification in information-seeking dialogues (Zamani et al., 2022) refers to the process of seeking

| Method | Shot | Prompt | Abg-CoQA | | | PACIFIC | | |
|---|---|---|---|---|---|---|---|---|
| | | | CNP | CQG | | CNP | CQG | |
| | | | F1 | BLEU-1 | Help. | F1 | ROUGE-2 | Help. |
| Baseline | - | - | 22.1 | 36.5 | 30.0 | 79.0 | 69.2 | 38.2 |
| SOTA | - | - | 23.6 | 38.2 | 56.0 | 86.9 | 90.7 | 80.1 |
| Vicuna-13B | 0 | Standard | - | 11.3 | 0.0 | - | 1.2 | 0.0 |
| | 1 | Standard | - | 11.4 | 0.0 | - | 2.5 | 0.0 |
| | 0 | Proactive | 4.1 | 13.2 | 0.0 | 2.3 | 2.3 | 0.0 |
| | 1 | Proactive | 12.1 | 13.2 | 4.5 | 0.0 | 3.3 | 0.0 |
| | 0 | ProCoT | 1.4 | 21.3 | 9.1 | 9.7 | 3.8 | 10.5 |
| | 1 | ProCoT | 18.3 | 23.7 | 22.7 | 27.0 | 41.3 | 33.1 |
| ChatGPT | 0 | Standard | - | 12.1 | 0.0 | - | 2.2 | 0.0 |
| | 1 | Standard | - | 12.3 | 0.0 | - | 2.0 | 0.0 |
| | 0 | Proactive | 22.0 | 13.7 | 17.6 | 19.4 | 2.9 | 0.0 |
| | 1 | Proactive | 20.4 | 23.4 | 23.5 | 17.7 | 14.0 | 12.5 |
| | 0 | ProCoT | 23.8 | 21.6 | 32.4 | 28.0 | 21.5 | 26.7 |
| | 1 | ProCoT | 27.9 | 18.4 | 45.9 | 27.7 | 16.2 | 35.8 |

Table 1: Experimental results on Abg-CoQA and PA-CIFIC datasets, whose baseline and SOTA results are adopting from Guo et al. (2021) and Deng et al. (2022a). **Bold** and underlined results denote the best performance for each LLM and the fine-tuned methods, respectively.

further information or details to better understand the topic or question at hand. In this context, clarification is an important part of the dialogue as it helps to ensure that the information being shared is accurate and complete.

#### 4.1.1 Problem Definition

Following previous studies (Aliannejadi et al., 2021; Guo et al., 2021; Deng et al., 2022a), the problem of asking clarification questions can be decomposed into two subtasks: 1) *Clarification Need Prediction* (**CNP**) to identify the necessity of clarification in the current turn, and 2) *Clarification Question Generation* (**CQG**) to produce an appropriate clarifying question if needed. Given the grounded document $\mathcal{D}$ and the dialogue context $\mathcal{C} = \{q_1, a_1, ..., q_{t-1}, a_{t-1}, q_t\}$, the dialogue system aims to first predict the binary ambiguity label $y$ on whether the current question $q_t$ needs to be clarified. If so, a corresponding clarification question should be generated as the response $a_t$ for clarifying the ambiguity.

#### 4.1.2 Experimental Setups

**Datasets.** We evaluate the capability of asking clarification questions in LLM-based dialogue systems on two types of datasets: 1) **Abg-CoQA** (Guo et al., 2021) in general domain, and 2) **PA-CIFIC** (Deng et al., 2022a) in finance domain. Details on these datasets can be found in Appendix A.

**Evaluation Metrics.** Following previous studies (Guo et al., 2021; Deng et al., 2022a), we use the F1 score for the evaluation of CNP, and BLEU-1 and ROUGE-2 (F1) for the evaluation of CQG. In addition, since the automatic lexical matching metrics may fail to actually estimate the clarification capability of the generated clarifying questions (Guo et al., 2021), we also adopt human evaluation to score whether the generated question is helpful for clarifying the existing ambiguity (**Help.**).

**Usage of LLMs.** To facilitate reproducibility, we adopt a static version of ChatGPT, *i.e.*, gpt-3.5-turbo-0301, and set the temperature to 0 for generating the deterministic outputs with the same inputs. In addition, we adopt an open-source LLM, *i.e.*, Vicuna-13B-delta-v1.1 , for the evaluation. The maximum number of new tokens is set to 128 for the generation.

**Prompting Schemes.** We evaluate the three prompting schemes introduced in Section 3, including standard, proactive, and ProCoT prompting. In addition, we report their results under both zero-shot and few-shot settings. Due to the limitation of the maximum sequence length in Vicuna (2,048 tokens), we only apply one-shot in-context learning for comparisons. The complete prompts adopted for evaluation is presented in Appendix C.

#### 4.1.3 Experimental Results

Table 1 summarizes the evaluation results on Abg-CoQA and PACIFIC datasets. There are several notable observations as follows:

**LLM-based conversational systems fail to ask clarification questions.** Under standard prompting, both Vicuna and ChatGPT fail to ask clarification questions when encountering ambiguous queries, according to the human evaluation on the helpfulness (**Help.**) of the generated responses for clarifying ambiguity. Even with one-shot demonstration, in-context learning (ICL) still cannot provide them with such ability. Under proactive prompting, given the option of clarification, Vicuna's ability to accurately take this action is still quite limited, with the **F1** scores close to 0. In contrast, ChatGPT becomes capable of asking clarification questions on Abg-CoQA, as evidenced by the improvement on both **F1** and **Help.** scores.

**ProCoT effectively endows LLM-based conversational systems with the capability of asking clarification questions.** Zero-shot ProCoT is not working in Vicuna, but one-shot ICL can largely improve the performance. As for Abg-CoQA, Chat-GPT with zero-shot ProCoT achieves competitive

| | Abg-CoQA | PACIFIC |
|---|---|---|
| Wrong Aspect | 21% | **30%** |
| Under-spec. Clari. | 16% | **23%** |
| Over-spec. Clari. | **15%** | 5% |
| Generation Error | **48%** | 42% |

Table 2: Statistics of error analysis.

performance with SOTA fine-tuned methods on the CNP task (**F1**), but the generated clarification questions are still unsatisfactory (**Help.**). One-shot ICL further improves the performance of ChatGPT with ProCoT to a great extent. The case study in Appendix D.1 shows that ProCoT also improves the explanability of asking clarification questions.

**As for domain-specific problem, there is still a noticeable gap from the fine-tuned methods.** Although ProCoT has already largely enhanced the capability of asking clarification questions, the performance of LLMs on the domain-specific task, *i.e.*, PACIFIC (Finance), is still far behind the fine-tuned methods. In fact, with fine-tuning on domain-specific data, the SOTA method can achieve a remarkable performance on PACIFIC, *i.e.*, 86.9 (**F1**) for CNP and 80.1 (**Help.**) for CQG, indicating the importance of domain knowledge.

### 4.1.4 Error Analysis

In order to find out the reason why LLM-based dialogue systems with ProCoT prompting fall short of handling domain-specific clarification dialogues, we randomly sample 100 error cases in clarification question generation from each dataset for analysis (all cases are generated by ChatGPT with one-shot ProCoT). We categorize these failure cases into four groups, including *Wrong Aspect*, *Under-specified Clarification*, *Over-specified Clarification*, and *Generation Error*. The details and examples can be found in the Appendix B. The statistics of error analysis is presented in Table 2. It can be observed that the proportion of failure cases attribute to the wrong aspect and under-specified clarification in PACIFIC (Finance) is higher than that in Abg-CoQA (General). This indicates that **ChatGPT may lack of certain domain knowledge required for asking precise and specific clarification questions.**

### 4.2 Target-guided Dialogues

Instead of making consistent responses to the user-oriented topics, the dialogue system for target-guided dialogues is required to proactively lead the conversation topics towards a designated tar-

get (Tang et al., 2019). According to different applications, the target can be topical keywords (Zhong et al., 2021), knowledge entities (Wu et al., 2019), or items to be recommended (Deng et al., 2023c).

### 4.2.1 Problem Definition

Given a target $\mathcal{D}$ that is only presented to the agent but unknown to the user, the dialogue starts from an arbitrary initial topic, and the system needs to produce multiple turns of responses $\{u_n\}$ to lead the conversation towards the target in the end. The produced responses should satisfy (i) **transition smoothness**, natural and appropriate content under the given dialogue context, and (ii) **target achievement**, driving the conversation towards the designated target. The problem is typically decomposed into two subtasks (Tang et al., 2019; Zhong et al., 2021; Yang et al., 2022): next topic selection and transition response generation.

### 4.2.2 Experimental Setups

**Datasets.** We first conduct turn-level evaluation of the target-guided capability on a next-turn target-oriented dataset **OTTers** (Sevegnani et al., 2021), which requires the dialogue system to proactively bridge the current conversation topic to approach the target. Furthermore, we adopt **TGConv** (Yang et al., 2022) to testify the ability to guide the multi-turn conversation to the target topic as the dialogue-level evaluation. Details can be found in Appendix A.

**Automatic Evaluation Metrics.** Following previous studies (Sevegnani et al., 2021; Yang et al., 2022), we adopt the hits@$k$ ($k \in [1, 3]$) for evaluating next topic prediction. Three text generation metrics, including BLEU, ROUGE-L, and METEOR scores, are used for the evaluation of response generation on the OTTers dataset.

As for the dialogue-level evaluation on the TGConv dataset, we follow existing studies (Yang et al., 2022; Wang et al., 2023b) to simulate multi-turn conversations via self-play (Tang et al., 2019), where the simulated user is unaware of the target topic. Three aspects are evaluated: 1) **Succ.** is the success rate of generating the target word within 8 turns of conversations; 2) **Turns** is the average turns of all dialogues that successfully reach the target word; and 3) **Coh.** is the contextual semantic similarity between the last utterance and the generated response, which is measured by MiniLM (Wang et al., 2020).

| Method | Shot | Prompt | Response Generation | | | Next Topic Prediction | |
|---|---|---|---|---|---|---|---|
| | | | BLEU | METEOR | R-L | hits@1 | hits@3 |
| GPT2 | - | - | 11.58 | 10.26 | 17.67 | 4.39 | 15.79 |
| DKRN | - | - | 12.86 | 11.90 | 21.52 | 4.91 | 17.72 |
| CKC | - | - | 13.34 | 11.65 | 24.77 | 6.87 | 21.89 |
| TopKG | - | - | 15.35 | 13.41 | 27.16 | 7.78 | 22.06 |
| Vicuna-13B | 0 | Standard | 10.01 | 13.27 | 16.00 | 12.01 | 19.03 |
| | 1 | Standard | 10.63 | 14.81 | 17.53 | 12.10 | 16.13 |
| | 0 | Proactive | 1.41 | 18.45 | 15.45 | 9.41 | 19.89 |
| | 1 | Proactive | **13.87** | **20.96** | **21.36** | 12.90 | **22.31** |
| | 0 | ProCoT | 5.27 | 16.59 | 15.96 | 11.56 | 18.01 |
| | 1 | ProCoT | 13.38 | 19.70 | 20.62 | **15.05** | 20.70 |
| ChatGPT | 0 | Standard | 11.34 | 20.62 | **18.26** | 13.44 | 27.69 |
| | 1 | Standard | 14.41 | 19.29 | 17.73 | 15.86 | 26.34 |
| | 0 | Proactive | 14.09 | **21.06** | 15.56 | 7.53 | 22.58 |
| | 1 | Proactive | **14.74** | 19.59 | 16.29 | 8.60 | 21.23 |
| | 0 | ProCoT | 10.20 | 19.57 | 15.97 | 12.63 | 23.92 |
| | 1 | ProCoT | 9.63 | 19.82 | 17.19 | **17.74** | **29.57** |

Table 3: Turn-level evaluation results on Next Topic Prediction and Transition Response Generation.

**Human Evaluation Metrics.** We also conduct the same human evaluation as Yang et al. (2022), including two dialogue-level metrics with the following instructions provided for annotators:

- Global-Coherence (G-Coh.): Whether the entire dialogue is logically and topically coherent.

- Effectiveness (Effect.): How efficiently the target is achieved.

A total of 100 dialogues are generated through simulation for each method. Three annotators assign ratings to the generated dialogues on a scale of [0, 1, 2], where higher scores indicate better quality.

**Baselines.** We report the results of several fine-tuned baselines for target-guided dialogues, including GPT-2 (Radford et al., 2019), DKRN (Qin et al., 2020), CKC (Zhong et al., 2021), TopKG (Yang et al., 2022), and COLOR (Wang et al., 2023b).

### 4.2.3 Turn-level Evaluation

Table 3 shows the turn-level evaluation results on OTTers. There are several notable observations:

**LLM-based dialogue systems are proficient at performing topic shifting towards the designated target.** According to the performance of LLMs with standard prompting, we observe that: 1) As for the next-topic prediction (**hits@k**), thanks to the extensive knowledge across various topics, zero-shot LLMs can achieve competitive (Vicuna) or even better (ChatGPT) performance than the fine-tuned methods. 2) As for the transition response generation, automatic evaluation metrics (**BLEU**, **METEOR**, **R-L**)[2] suggest that zero-shot

---
[2]Note that the automatic evaluation of response generation is less reliable (Sevegnani et al., 2021), as the same topic

models perform closely to fine-tuned methods in terms of lexical similarity with the reference response. 3) One-shot ICL casts no positive impact on the performance and may even lead to worse results in next-topic prediction. This indicates that it is difficult for LLMs to enhance the topic shifting capability from limited demonstrations.

**Only ProCoT prompting with one-shot demonstrations can improve the topic shifting capability.** Without demonstrations, proactive and ProCoT prompts perform even worse than standard prompts, since LLMs may confuse about what kinds of topics are desired. For example, we observe a typical mistake that LLMs tend to analyse the next topics using questions, such as "*What kind of food do you like?*", leading to a narrow topic for the next turn. With one-shot demonstrations, ChatGPT with proactive prompts continues to underperform compared to standard prompts when it comes to accurately predicting suitable topics towards the target. However, it is worth noting that only ProCoT prompts consistently show an improvement in the performance of all LLMs for next topic prediction.

### 4.2.4 Dialogue-level Evaluation

Table 4 shows the dialogue-level evaluation results on TGConv. We draw the following conclusions:

**LLM-based dialogue systems tend to make aggressive topic transition.** The results demonstrate the effectiveness of LLMs in steering the conversation towards the designated target, with ChatGPT exhibiting nearly perfect success rates (**Succ.**). Compared with baselines, LLMs also excel in generating more coherent responses that align with the dialogue context (**Coh.**), showcasing their impressive abilities in context understanding and response generation. Furthermore, the analysis reveals that ChatGPT basically achieves the target topics within just three turns, suggesting its tendency to generate responses that aggressively involve the desired topic. Similar observations can be made with Vicuna using standard prompting.

**ProCoT prompting enables a smoother topic transition of target-guided dialogues.** Under proactive prompting, the response coherency is improved by the topic planning. However, the success rate is negatively affected, which attributes to

---
can be described in different ways rather than the reference response. We mainly discuss the topic shifting capability in terms of the performance on next topic prediction.

| Method | Shot | Prompt | Easy Target | | | Hard Target | | |
|---|---|---|---|---|---|---|---|---|
| | | | Succ.(%) | Turns | Coh. | Succ.(%) | Turns | Coh. |
| GPT2 | - | - | 22.3 | 2.86 | 0.23 | 17.3 | 2.94 | 0.21 |
| DKRN | - | - | 38.6 | 4.24 | 0.33 | 21.7 | 7.19 | 0.31 |
| CKC | - | - | 41.9 | 4.08 | 0.35 | 24.8 | 6.88 | 0.33 |
| TopKG | - | - | 48.9 | 3.95 | 0.31 | 27.3 | 4.96 | 0.33 |
| COLOR | - | - | 66.3 | - | 0.36 | 30.1 | - | 0.35 |
| Vicuna-13B | 0 | Standard | 63.0 | **2.63** | 0.43 | 62.5 | **2.45** | 0.39 |
| | 1 | Standard | 62.7 | 2.83 | 0.45 | **65.0** | 2.90 | 0.43 |
| | 0 | Proactive | 37.8 | 2.71 | 0.48 | 35.6 | 2.56 | **0.55** |
| | 1 | Proactive | 48.3 | 2.71 | 0.50 | 34.6 | 2.95 | 0.51 |
| | 0 | ProCoT | 65.2 | 4.22 | 0.49 | 54.9 | 4.17 | 0.45 |
| | 1 | ProCoT | **72.3** | 3.55 | **0.52** | 59.8 | 3.81 | 0.48 |
| ChatGPT | 0 | Standard | **97.5** | **2.26** | 0.38 | **96.3** | 2.30 | 0.41 |
| | 1 | Standard | 96.3 | 2.42 | 0.42 | 93.5 | **2.28** | 0.38 |
| | 0 | Proactive | 85.9 | 3.20 | **0.47** | 83.0 | 2.83 | **0.43** |
| | 1 | Proactive | 90.7 | 2.86 | 0.36 | 86.2 | 2.94 | 0.31 |
| | 0 | ProCoT | 96.3 | 2.47 | 0.41 | 92.0 | 2.29 | 0.34 |
| | 1 | ProCoT | 95.9 | 2.63 | 0.45 | 92.1 | 2.47 | 0.39 |

Table 4: Dialogue-level evaluation results on target-guided dialogues.

| Method | Prompt | Easy Target | | Hard Target | |
|---|---|---|---|---|---|
| | | G-Coh. | Effect. | G-Coh. | Effect. |
| TopKG | - | 1.42 | 1.24 | 1.21 | 1.10 |
| Vicuna-13B | Standard | 1.37 | 1.60 | 1.20 | 1.49 |
| | Proactive | 1.51 | 1.27 | 1.26 | 1.23 |
| | ProCoT | **1.57** | **1.70** | **1.35** | **1.59** |
| ChatGPT | Standard | 0.97 | **1.92** | 0.84 | **1.89** |
| | Proactive | **1.24** | 1.77 | 1.12 | 1.68 |
| | ProCoT | 1.20 | 1.90 | **1.14** | 1.85 |

Table 5: Human evaluation on target-guided dialogues. All reported methods are under the one-shot setting.

its drawback of next topic prediction discussed in Section 4.2.3. Under ProCoT prompting, Vicuna effectively guide the conversation towards the designated target with a smoother (higher **Coh.**) and more engaging (higher **Turns**) conversation than using standard prompting. However, it still remains challenging for ChatGPT to perform a smooth topic transition. Case studies in Appendix D.2 provide intuitive examples for illustrating these observations.

### 4.2.5 Human Evaluation

Table 5 presents the human evaluation results on TGConv. Compared with TopKG, LLMs demonstrate remarkable efficiency in achieving the designated target (**Effect.**). However, the global coherence (**G-Coh.**) of the generated dialogues by ChatGPT is quite low, which may harm the user engagement and experience during the conversation. Thus, the proficiency of controllable generation in LLMs is a double-edged sword for target-guided dialogues. **The key challenge of LLMs is how to guarantee the topical smoothness and coherence of the generated transition responses.**

| Method | Shot | Prompt | Nego. Strategy | | Dial. Act | | Resp. Gen. | |
|---|---|---|---|---|---|---|---|---|
| | | | F1 | AUC | F1 | AUC | BLEU | BERTScore |
| FeHED | - | - | 17.6 | 55.8 | 20.6 | 76.9 | 23.7 | 27.0 |
| HED+RNN | - | - | 23.2 | 65.3 | 33.0 | 83.1 | 22.5 | 22.8 |
| HED+TFM | - | - | 26.3 | 68.2 | 32.5 | 85.6 | 24.4 | 27.7 |
| DIALOGRAPH | - | - | 26.1 | 68.1 | 33.4 | 85.6 | 24.7 | 28.1 |
| Vicuna-13B | 0 | Standard | - | - | - | - | 1.7 | -14.0 |
| | 1 | Standard | - | - | - | - | 1.9 | -2.8 |
| | 0 | Proactive | 20.6 | 51.1 | 4.2 | 50.3 | 2.3 | -7.0 |
| | 1 | Proactive | 15.2 | 50.0 | 6.7 | 50.8 | 2.6 | -0.9 |
| | 0 | ProCoT | 19.0 | 49.7 | 3.6 | 50.3 | 2.6 | -6.2 |
| | 1 | ProCoT | 17.8 | 48.9 | **7.7** | **52.5** | 2.6 | **-0.9** |
| ChatGPT | 0 | Standard | - | - | - | - | 2.3 | -4.3 |
| | 1 | Standard | - | - | - | - | 3.1 | 0.7 |
| | 0 | Proactive | 12.8 | 51.3 | 13.3 | 56.3 | **4.2** | 1.3 |
| | 1 | Proactive | 13.7 | 50.9 | 12.0 | 54.9 | 3.9 | **2.9** |
| | 0 | ProCoT | 10.8 | 50.4 | 10.1 | 54.2 | 3.7 | -0.9 |
| | 1 | ProCoT | **15.1** | **55.5** | **16.3** | **58.2** | 3.9 | 1.6 |

Table 6: Evaluation results on Negotiation Strategy Prediction, Dialogue Act Prediction, and Response Generation.

### 4.3 Non-collaborative Dialogues

Unlike collaborative task-oriented dialogue settings (Zhang et al., 2020c), where the user and the system work together to reach a common goal (*e.g.*, booking hotels), in non-collaborative dialogues, the user and the system have a conflict of interest but aim to strategically communicate to reach an agreement (*e.g.*, negotiation) (Zhan et al., 2022). The system is required to leverage a series of proactive strategies to reach an agreement favorable to itself, instead of passively following the user's intents.

### 4.3.1 Problem Definition

Given the dialogue history $\mathcal{C} = \{u_1, ..., u_{t-1}\}$ and the dialogue background $\mathcal{D}$, the goal is to generate a response $u_t$ with appropriate dialogue strategy $a_t$ that can lead to a consensus between the system and user. A set of dialogue strategies $\mathcal{A}$ is pre-defined for prediction. Based on different applications, the dialogue strategy can be coarse dialogue act labels or fine-grained strategy labels. The dialogue background includes the system's goal and the related grounded information, such as item descriptions in bargain negotiation (He et al., 2018) and user profile in persuasion dialogues (Wang et al., 2019).

### 4.3.2 Experimental Setups

**Datasets.** We use the **CraigslistBargain** dataset (He et al., 2018) for evaluating the capability of strategically handling non-collaboration in LLM-based dialogue systems. The dataset was created under the bargain negotiation setting where the buyer and the seller are negotiating the price of an item on sale. Details can be found in Appendix A.

| Metric | Standard | Proactive | ProCoT | Gold |
|---|---|---|---|---|
| Persuasive | 1.24 | 1.28 | 1.43 | **1.54** |
| Coherent | 1.56 | 1.66 | **1.74** | 1.69 |
| Natural | 1.94 | 1.82 | 1.89 | **1.97** |
| Win Rates | | | | |
| - vs. Standard | - | 0.22 | 0.24 | **0.42** |
| - vs. Proactive | 0.25 | - | 0.31 | **0.45** |
| - vs. ProCoT | 0.20 | 0.18 | - | **0.34** |
| - vs. Gold | 0.19 | 0.09 | 0.23 | - |
| Sale-to-List Ratio | 0.48 | 0.43 | 0.54 | **0.64** |

Table 7: Human evaluation on non-collaborative dialogues. All reported methods are based on ChatGPT under the one-shot setting. **Gold** denotes that we instruct the LLMs to generate responses conditioned on the reference dialogue acts and negotiation strategies.

**Automatic Evaluation Metrics.** Following the previous study (Joshi et al., 2021), we conduct a comprehensive evaluation over three subtasks, including negotiation strategy prediction, dialogue act prediction, and response generation. We report the F1 and ROC AUC scores for strategy prediction and dialogue act prediction, where the former one is a multi-label prediction problem. For the response generation, we adopt BLEU score and BERTScore (Zhang et al., 2020a) for evaluation.

**Human Evaluation Metrics.** Following Joshi et al. (2021), we also conduct human evaluation on 100 randomly sampled dialogues with both subjective and objective human judgement. As for the subjective judgement, annotators are asked to score [0,1,2] on how persuasive, coherent, and natural the generated response is.

We further pair the generated responses from each prompting scheme, including Standard, Proactive, ProCoT, and Ground-truth (GT), with the corresponding responses from each of the other prompting scheme to compute the overall win rates between each pair.

As for the objective human judgement, we adopt the sale-to-list ratio (SL%) (Joshi et al., 2021; Dutt et al., 2021) as an indicator for explicitly measuring the negotiation inclination in the generated response:

$$SL\% = \frac{\text{bargain price} - \text{buyer target price}}{\text{listed price} - \text{buyer target price}}, \quad (4)$$

where the bargain price is the price that the seller would like to sell the item at the current turn. The lower the SL%, the more compromise the seller have made.

To sum up, the instructions provided for annotators are as follows:

- Persuasive: Whether the seller is persuasive in bargaining the price.

- Coherent: Whether the seller's responses are on topic and in line with the conversation history.

- Natural: Whether the seller is human-like.

- Bargain Price: What is the current bargain price from the seller's side.

- Win: Assume you are the seller. Which dialogue system you would like to use for bargain the price with the buyer (Win/Tie/Lose).

**Usage of LLMs & Prompting Schemes.** The adopted LLMs are the same, but the maximum number of new tokens is set to be 256, as there are more information needed to be generated, including negotiation strategies and dialogue acts.

**Baselines.** We compare several fine-tuned SOTA baselines for negotiation dialogues, including Fe-HED (Zhou et al., 2020), HED+RNN/TFM, and DIALOGRAPH (Joshi et al., 2021).

### 4.3.3 Experimental Results

Table 6 and Table 7 present the results with automatic and human evaluation metrics, respectively. There are several notable findings as follows:

**LLM-based dialogue systems fail to predict appropriate negotiation strategies and dialogue acts.** Table 6 shows that failures on strategy learning further result in a poor performance of response generation. Specifically, ChatGPT generally performs better than Vicuna in strategy learning. Although both proactive and ProCoT prompting schemes can slightly improve the final performance of response generation, there is still a large gap from fine-tuned methods according to automatic evaluation metrics.

**The key challenge of LLMs in handling non-collaborative dialogues is how to effectively optimize the strategy planning.** Table 7 shows that the generated responses conditioned on reference strategies are more favorable (**Win Rates**). In specific, ChatGPT guarantees a high score on the human-like response generation (**Natural**). With the ProCoT, the generated responses are more coherent to the conversation history (**Coherent**), which can also be observed from the case study in

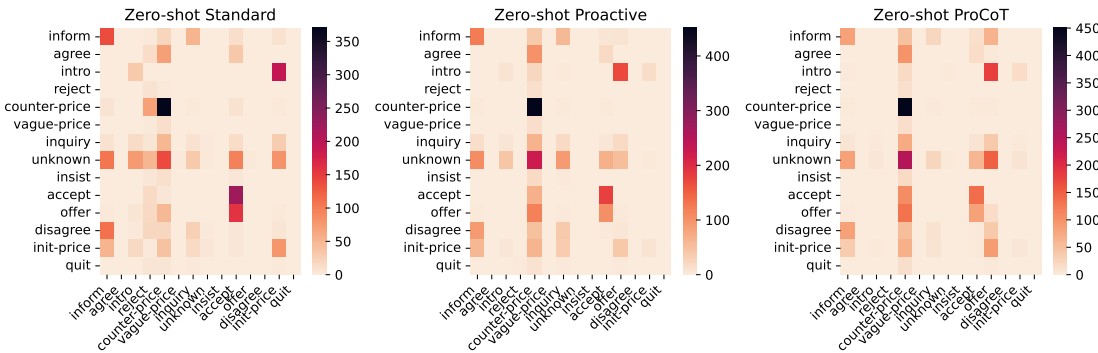

Figure 2: Heatmaps on the relationships between target and predicted dialogue acts. As no dialogue act is predicted in standard prompting, a dialogue act classifier is trained to identify the dialogue act of the generated response.

Appendix D.3. However, compared with prompting with reference strategies, all the other prompting schemes fall short of generating persuasive responses for negotiation (**Persuasive**), indicating their shortcomings on strategy learning. This is also validated by the objective judgement on **Sale-to-List Ratio**, which shows that ChatGPT can reach a better deal for itself when being conditioned on reference strategies. Similarly, Chen et al. (2023c) empirically show that, given the optimal planned strategy, ChatGPT achieves strong performance on controllable response generation in some other strategy-based dialogues.

### 4.3.4 Analysis of Strategy Learning

Figure 2 presents the analysis of the relationships between the target and predicted dialogue acts by ChatGPT. As for the standard prompting, we observe two typical mistakes: 1) The system tends to propose the initial bargain price (`init-price`), instead of greetings (`intro`) and waiting for the buyer to initialize the bargain. 2) The system often directly accepts the buyer's offer (`accept`) when it is supposed to offer another price for negotiation (`offer`). This also explains why the **Sale-to-List Ratio** is relatively low when using standard prompting in Table 7. On the other hand, Proactive and ProCoT prompting share similar patterns of mistakes, where ChatGPT tends to propose a counter price (`counter-price`) to negotiate with the buyer.

Appendix E presents the analysis of the distribution of selected strategies by ChatGPT. In the reference responses, the seller often shows positive/negative sentiment to negotiate with the buyer. However, ChatGPT inclines to adopt conservative or concessionary strategies, such as using hedge words, show gratitude, or propose a counter price. Overall, we conclude that **ChatGPT tends to make compromise with the buyer during the**

**negotiation, rather than strategically taking actions to maximize its own benefit**.

## 5 Conclusion

In this work, we conduct the first comprehensive evaluation on the capability of LLM-based dialogue systems in handling proactive dialogues, including clarification, target-guided, and non-collaborative dialogues. To enhance the proactivity of LLM-based dialogue systems, we propose a proactive chain-of-thought prompting scheme that triggers the reasoning and planning capability of LLMs. The empirical analysis sheds light on the potentials of LLMs for proactive dialogues: 1) ProCoT largely enhances the originally poor performance of LLMs in asking clarification questions, but still limits in handling domain-specific applications. 2) LLM-based dialogue systems perform aggressive topic shifting towards the designated target, while ProCoT enables the topic planning to be smoother. 3) Despite the strength on controllable response generation, the capability of strategy learning and planning is a key challenge for LLMs in handling non-collaborative dialogues.

## Acknowledgement

This research is supported by NExT Research Center.

## Limitation

In this section, we discuss the limitations of this work from the following perspectives:

**Sensitivity of Prompts**   Similar to other studies on prompting LLMs for dialogue applications (Lee et al., 2023; Chen et al., 2023c,a), the evaluation results are likely to be sensitive to the choice of prompts. Besides, it is also likely that the designed prompts are not the optimal ones for the concerned

problem. In fact, prompt sensitivity and optimality themselves are valuable research problems in dialogue systems, which can be further investigated in the future studies. To facilitate the reproducibility of this work, we will release all the prompts used in the experiments and provide detailed descriptions about the designs of each prompting scheme in Appendix C. The code and data will be released via `https://github.com/dengyang17/LLM-Proactive`.

**Financial and Computational Cost of LLMs** It is financially expensive to call the API of commercial LLMs for experiments. In our experiments, it costs about $120 to call the OpenAI API for getting all the experimental results of ChatGPT. On the other hand, it is computationally expensive to conduct experiments with open-source LLMs in local machines. In our experiments, we choose Vicuna 13B as the open-source LLM for evaluation, which can be adapted to NVIDIA DGX-1 V100 32G for inference. If more budgets and better experimental environment are permitted, it would be great to evaluate how other larger LLMs performs in the concerned proactive dialogue problems, such as GPT-4, LLaMA/Vicuna 65B, etc.

**Capability of Planning and Decision Making** The proposed ProCoT prompting scheme can be regarded as a preliminary attempt at triggering the capability of planning and decision making from LLM-based dialogue systems. Compared with finetuned methods, such ability of LLMs is still weak as we learn from the empirical analysis. Moreover, simply prompting LLMs to be proactive may fall short of handling decision making under dynamic environments in real-world applications. It is worth studying how LLM-based dialogue systems handle the proactive dialogue problems in an interactive setting with more diverse user simulation (Lei et al., 2022; Fu et al., 2023).

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

## A  Details of Datasets

In the experiment, we adopt the test sets from five datasets for evaluation, including Abg-CoQA (Guo et al., 2021), PACIFIC (Deng et al., 2022a), Otters (Sevegnani et al., 2021), TGConv (Yang et al., 2022), and CraigslistBargain (He et al., 2018). Detailed descriptions of each dataset are as follows:

- Abg-CoQA[3] is constructed based on the CoQA dataset (Reddy et al., 2019) by truncating a partial conversation from the full conversation and selecting ambiguous questions.

- PACIFIC[4] is constructed based on the TAT-QA dataset (Zhu et al., 2021), an question answering dataset in the financial domain, whose contexts contain a hybrid of tables and texts. Deng et al. (2022a) rewrite the questions to be ambiguous for introducing clarification turns in the conversation.

- OTTers is a next-turn target-oriented dialogue dataset, which requires the agent proactively generate a transition utterance to approach the designated target. We adopt the processed version[5] by Yang et al. (2022) for evaluation. The topic is represented as a set of topical keywords.

- TGConv is constructed based on ConvAI2 (Dinan et al., 2019) and is split to two settings, including "easy-to-reach" and "hard-to-reach". The topic is also represented as a set of topical keywords.

- CraigslistBargain was created in a negotiation setting where two crowdsourced workers play the roles of the buyer and the seller to bargain the price of an item. We adopt the processed version[6] by Joshi et al. (2021) for evaluation, which assigns 10 dialogue acts and 21 negotiation strategies to the utterances.

## B  Error Analysis Details for Clarification Dialogues

As shown in Table 9, we categorize these failure cases into the following four groups:

---

[3] https://github.com/MeiqiGuo/AKBC2021-Abg-CoQA/tree/main/abg-coqa

[4] https://github.com/dengyang17/PACIFIC/tree/main/data/pacific. Since the labels in the test set is not publicly released, we adopt the validation set for evaluation.

[5] https://github.com/yyyyyyyzt/topkgchat

[6] https://github.com/rishabhjoshi/DialoGraph_ICLR21/tree/main

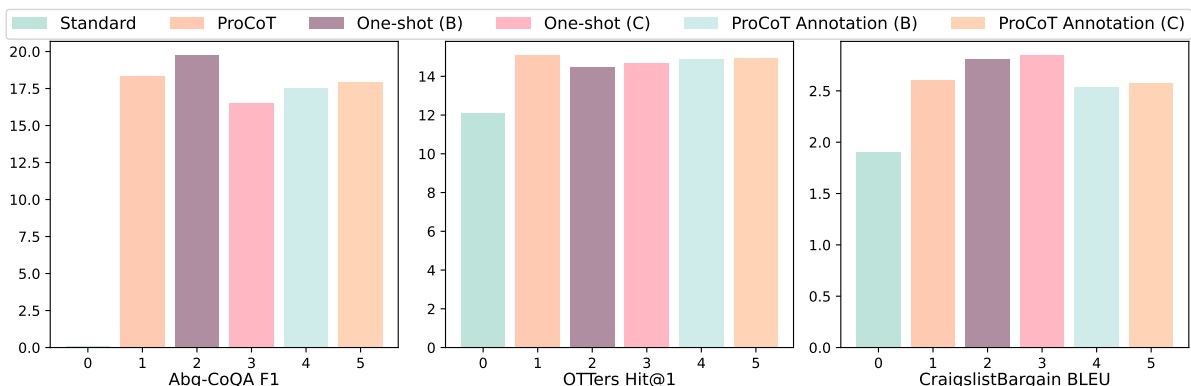

Figure 3: Performance in terms of different prompts, including different one-shot examples and different ProCoT annotations. The reported results are based on Vicuna-13B.

| Dataset | #Dialog | Avg. #Turns | Avg. #Words |
|---|---|---|---|
| Abg-CoQA | 1055 | 5.04 | 4.87 |
| PACIFIC | 278 | 7.10 | 6.80 |
| OTTers | 372 | 3.00 | 9.19 |
| TGConv | 751 | 8.97 | 12.47 |
| CraigslistBargain | 567 | 9.78 | 11.65 |

Table 8: Statistics of datasets.

| Under-specified Clarification | Q: What is the 2019 average defined schemes? G: What kind of defined schemes are you asking about? P: Could you please clarify what you mean by "average defined schemes"? |
|---|---|
| Wrong Aspect | Q: What was the increase / (decrease) in the cost? G: Which period are you asking about? P: Which cost are you referring to? |
| Over-specified Clarification | Q: How much amount of goodwill acquisitions was done in 2019? G: What kind of goodwill acquisitions are you asking about? P: Are you asking about the total amount of goodwill activity or specifically about acquisitions in 2019? |
| Generation Error | Q: How much was the average operating income? G: Which period are you asking about? P: The document does not provide information on the average operating income. Therefore, the question cannot be answered. |

Table 9: Error Analysis (Q: User Query, G: Ground-truth, P: Prediction).

- *Wrong Aspect*: The model generates a question for clarifying a wrong aspect of the user query.

- *Under-specified Clarification*: The model generates an under-specified clarification question, where the requested information is too general so that it would be difficult for the user to provide feedbacks.

- *Over-specified Clarification*: The model generates an over-specified clarification question, where the requested information is already clear in the user query.

- *Generation Error*: Although the model identifies the need for clarification, but it doesn't generate the output as the required format, such as no clarification question.

## C  Designs of Prompts

Table 10, 11, and 12 present the example of prompts for clarification, target-guided, non-collaborative dialogues, respectively. As for the zero-shot setting, the overall prompt is composed by the task instruction and the sample. As for the few-shot setting, the overall prompt is composed by the task instruction, a number of samples with demonstrations, and the test sample.

In particular, we clarify several questions regarding the prompt designs as follows:

**How to construct the task instructions?**  The task instructions first follow the problem definition for each proactive dialogue problem. Then, similar to other studies on applying LLMs for different tasks (Wei et al., 2023; Bang et al., 2023), we further instruct the LLMs to generate the response following the desired output format for evaluation.

## Clarification Dialogues: **Abg-CoQA**

**Task Background**: The grounded document is "Santa Claus makes sure that every year at Christmas time that his reindeer are ready to deliver gifts to girls and boys all over the world. Without the reindeer, how would Santa fly from home to home? The most important thing that Santa has to do for the reindeer is to make sure they have their coffee on Christmas Eve. Without the coffee, they won't stay awake for very long. Sleeping reindeer are not very helpful. Santa also gives the reindeer candy to keep up their energy. In the North Pole, the main candy is candy canes in red and white. Reindeer love the candy canes that are different colors best and Santa says that helps them fly faster. Next, Santa has to make sure that the reindeer have a good meal before they go. Santa wants to make sure he doesn't have to share the cookies kids leave for him. The last thing Santa has to do is make sure the reindeer know where they are going. Santa has only been lost one time on Christmas Eve and he does not want that to happen again."

**Conversation History**: ["User": "What candy does he feed them most often?", "System": "Candy canes.", "User": "How many times has he been lost?", "System": "Once.", "User": "Why does he make sure they are well fed?"]

**Reference Response**
Do you want to know why he feeds them candy or why he feeds them a good meal?
A: Why he feeds them candy — to keep up their energy
B: Why he feeds them a good meal — to make sure he doesn't have to share the cookies kids leave for him

**Generated Responses**
**(a) Standard Prompting (ChatGPT)**: To ensure that they have enough energy for their long journey delivering gifts.
**(b) Proactive Prompting (ChatGPT)**: The clarifying question is "Who are you referring to when you say 'he'?"
**(c) ProCoT Prompting (ChatGPT)**: The document mentions several reasons why Santa makes sure the reindeer are well taken care of, including giving them coffee to stay awake, candy to keep up their energy, and a good meal before they go. Therefore, the question is ambiguous. The clarifying question is "Which reason are you referring to?"

Figure 4: Case study from the Abg-CoQA dataset.

**How to choose the one-shot sample?** Due to the input length limitation of LLMs, we could only adopt one-shot in-context learning (ICL). In order to testify the sensitivity of the choice of the one-shot sample, we report the results with three different one-shot samples in Figure 3. Despite the variance among different one-shot examples as expected when using exemplar-based ICL (Scao and Rush, 2021; Zhao et al., 2021), the observations of adopting ProCoT for different proactive dialogues remain unchanged. Since the variance of one-shot ICL is inevitable, we simply adopt the first dialogue sample in the original training set of each dataset as the one-shot sample for facilitating reproducibility.

**How to construct the demonstration of proactive chain-of-thoughts?** The demonstration of

## Target-guided Dialogues: **OTTers**

**Task Background**: The target topic is ["piano", "listen"]
**Conversation History**: ["User": "I really love dogs."]

**Reference Transition Response and Next Topics**
A: My dogs fall asleep to music. ["dog", "fall", "music"]
B: I love dogs too and I also love music. ["love", "dog", "music"]
C: My dog calms down when listening to music. ["dog", "music", "calm", "listen"]

**Generated Responses**
**(a) Standard Prompting (ChatGPT)**: Dogs are amazing companions, aren't they? Have you ever tried listening to music while playing the piano? It can be a great way to relax and unwind.
**(b) Proactive Prompting (ChatGPT)**: The next topics are ["pets", "music", "favorite"]. The response is "Pets and music are two of my favorite things. Do you have a favorite type of pet or music to listen to?"
**(c) ProCoT Prompting (ChatGPT)**: The current topic is ["dogs", "love"]. To bridge the current topic with the target topics, the next topics are ["pets", "music", "hobbies", "passion"]. Based on the predicted next topics, the response is "Pets can bring so much joy and love into our lives. Speaking of hobbies and passions, do you enjoy listening to music or playing any instruments like the piano?"

Figure 5: Case study from the OTTers dataset.

proactive chain-of-thoughts is written by human annotators, which represents their own chain-of-thoughts of the planning or decision making for the proactive dialogues. Following Wei et al. (2022b), we also report the results with the demonstrations from three different annotations. As shown in Figure 3, it can be observed that the variance is much smaller than the one-shot sample. We adopt the best ProCoT annotation for each dataset in the evaluation.

## D Case Study

In order to intuitively compare the three prompting schemes, we conduct case studies on the generated responses regarding three proactive dialogue problems.

### D.1 Clarification Dialogues

Figure 4 presents a case study from the Abg-CoQA dataset. There are two possible answers for the ambiguous user question in this case. When using standard prompting (a), ChatGPT generates a response based on a random guess, potentially resulting in an answer that may not align with the user's desired outcome. When using proactive prompting (b), although ChatGPT decides to ask a clarification question, the concerned aspect of clarification is not the focus of this conversation. ProCoT prompt-

**Target-guided Dialogues: TGConv**

**Task Background**: The target topic is "Gaullist"
**Conversation History**: ["User": "I don't enjoy wine but I do enjoy psychology journals"]

---

**Self-play Dialogues**

**(a) Standard Prompting (Vicuna-13B)**
**User**: I don't enjoy wine but I do enjoy psychology journals
**System**: That's interesting. Have you ever read any articles on the psychology of wine consumption?
**User**: I haven't read any articles on the psychology of wine consumption specifically, but I have read some articles on the psychology of food and drink consumption in general.
**System**: I'm glad you're interested in the psychology of wine consumption.
…
*Failed to reach the target in 8 turns.*

**(b) Proactive Prompting (Vicuna-13B)**
**User**: I don't enjoy wine but I do enjoy psychology journals
**System**: I do enjoy reading, especially psychology journals
**User**: That's great to hear! What's your favorite psychology journal to read?
**System**: My favorite psychology journal to read is the journal of personality and social psychology
…
*Failed to reach the target in 8 turns.*

**(c) ProCoT Prompting (Vicuna-13B)**
**User**: I don't enjoy wine but I do enjoy psychology journals
**System**: I am interested in psychology and reading about it in journals
**User**: That's great to hear! There are many interesting psychology journals out there. Have you read any particularly good ones lately?
**System**: I have been reading a lot of **gaullist** psychology journals lately, they have some really interesting studies on the effects of wine on the mind.
*Successful to reach the target in 4 turns!*

**(d) ProCoT Prompting (ChatGPT)**
**User**: I don't enjoy wine but I do enjoy psychology journals
**System**: Interesting, have you ever read any **Gaullist** political philosophy? It's a fascinating topic that combines both politics and psychology.
*Successful to reach the target in 2 turns!*

Figure 6: Case study from the TGConv dataset.

ing (c) first provides a thorough analysis to identify the underlying reasons for the ambiguity in the user's question. This analysis serves as the foundation for generating an effective clarifying question, which helps to clarify and disambiguate the user's intended meaning.

## D.2 Target-guided Dialogues

Figure 5 presents a case study from the OTTers dataset, where the target topical keywords include "piano" and "listen", and the system is required to generate a transition response to approach the target topics. It can be observed that the current topics in the user utterance, *i.e.,* "dog", is completely different from the target topics. When using standard

---

**Non-collaborative Dialogues: CraigslistBargain**

**Task Background**: The item description is "Furniture. It's a solid wood tan color computer table with swivel chair for sale. It's in good condition as shown in pics. Moving out sale computer table and chair". The target selling price is 50.

---

**Turn 3**
**Conversation History**: ['Seller': 'Hi there, how are you today?', 'Buyer': 'What are the dimensions?']

**Reference**: 40 by 60 inches. Fairly large, in good condition.
**Standard (ChatGPT)**: The dimensions are [insert dimensions here]. Are you interested in purchasing it for the listed price of $50 or were you hoping to negotiate a better deal?
**Proactive (ChatGPT)**: The most appropriate set of negotiation strategies is ['informal', 'greet', 'inform'] and the most appropriate dialogue act is ['inquiry']. The response is "The dimensions are [insert dimensions here]. Is there anything else you would like to know?"
**ProCoT (ChatGPT)**: The negotiation has not yet started, as the buyer is asking for more information about the item. The appropriate goal is to provide the requested information and engage the buyer in the negotiation. To reach this goal, the most appropriate set of negotiation strategies is ['informal', 'greet', 'inform'] and the most appropriate dialogue act is ['inquiry']. Based on the selected negotiation strategies and dialogue act, the response is "Hi, I'm doing well, thank you for asking. The dimensions of the computer table are [insert dimensions here]. Is there anything else you would like to know about it?"

**Turn 9**
**Conversation History**: ['Seller': 'Hi there, how are you today?', 'Buyer': 'What are the dimensions?', 'Seller': '40 by 60 inches. Fairly large, in good condition.', 'Buyer': 'Does the chair have rollers?', 'Seller': 'yes, it does', 'Buyer': 'It's bigger than I had wanted but might be able to make it work. Would you take $30?', 'Seller': 'I would be wiling to do $40 with delivery included.', 'Buyer': 'I can pick it up but thanks for offering. Best I could do is $32']

**Reference**: $35 and you have a deal.
**Standard (ChatGPT)**: I understand your budget constraints, but I was hoping to get closer to my target selling price of $50. Would you be willing to meet me halfway at $45?
**Proactive (ChatGPT)**: The most appropriate set of negotiation strategies is ['hedge', 'informal', 'gratitude', 'neg_sentiment'] and the most appropriate dialogue act is ['counter-price']. The response is "I appreciate your offer, but the lowest I can go is $45. It's already a great deal with the swivel chair included."
**ProCoT (ChatGPT)**: The buyer has proposed a price that is lower than the target selling price. The seller has already offered a counter-price with delivery included, but the buyer is still negotiating. The next step should be to politely reject the buyer's offer and propose a new counter-price closer to the target selling price. To reach this goal, the most appropriate set of negotiation strategies is ['please', 'certainty', 'propose', 'hedge', 'gratitude'] and the most appropriate dialogue act is ['counter-price']. Based on the selected negotiation strategies and dialogue act, the response is "I appreciate your offer, but I cannot go lower than $38. Would that work for you?"

Figure 7: Case study from the CraigslistBargain dataset.

prompting, ChatGPT leverages its overwhelming ability of controllable generation to generate the response with aggressive topic transition. Despite its fluency, it is not a natural utterance with log-

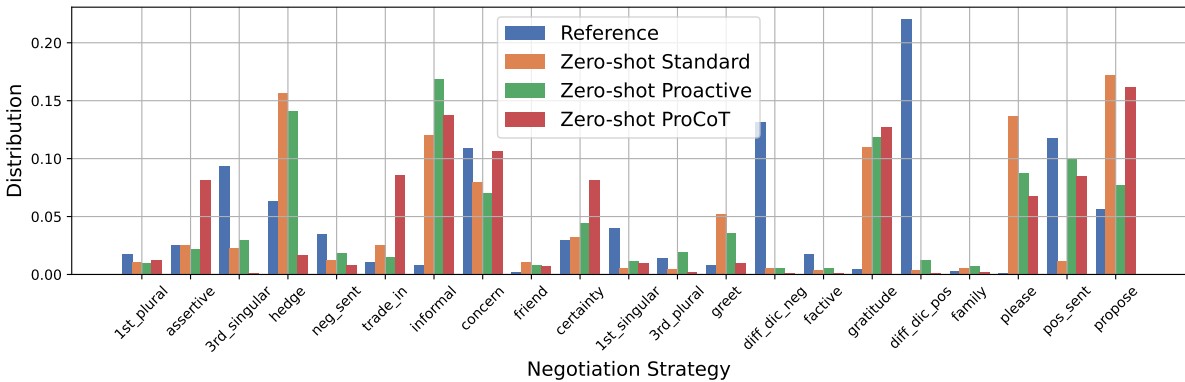

Figure 8: Distribution of selected negotiation strategies. Similarly, a negotiation strategy classifier is trained to identify the negotiation strategies of the generated response in standard prompting.

ical or topical coherency. When using proactive prompting, although the predicted next topics are less aggressive than the standard prompting, the generated transition response just blends the current topics and the next topics together without naturally connecting the topics. Finally, ChatGPT with the ProCoT prompting generates a relatively smoother transition response to bridge the current topic with the target topic through planning about the topic shifting process.

Figure 6 presents a case study from the TGConv dataset, where the hard-to-reach target of this case is "Gaullist", and the system is required to reach this target within 8 turns of conversations under the self-play simulation. As for Vicuna, it is struggled to lead the conversation towards this hard-to-reach target, when using standard and proactive prompting. ProCoT prompting enables Vicuna to effectively and smoothly drive the conversation towards the designated target. In addition, owing to the powerful capability of controllable text generation, ChatGPT directly responds with the target topic to the initial user utterance. However, the topic transition is relatively aggressive, which might downgrade the user engagement or experience during the conversation.

### D.3 Non-collaborative Dialogues

Figure 7 presents a case study from the Craigslist-Bargain dataset, where the system plays the seller role to bargain with the buyer. At turn 3, even though the buyer just inquires about the item information without showing the bargain intention, ChatGPT with standard prompting tends to initiate the negotiation, which may put the seller in a disadvantageous position. Proactive and ProCoT

prompting enable the dialogue act and strategy prediction of the next response. Especially for the analysis of the current negotiation status, ProCoT points out that the negotiation has not yet started.

At turn 9, we observe that the seller has already lowered down the bargain price to $40 in a previous turn. Without the reasoning and planning process, ChatGPT with standard and proactive prompting generates the response with contradictory statement, *i.e.,* propose a higher counter price ($45) for bargain, which is unreasonable in negotiation dialogues. With proactive CoTs, ChatGPT effectively summarizes the current negotiation progress and makes a better decision on the next negotiation goal.

### E Analysis of Strategy Learning (Cont.)

Figure 8 presents the analysis of the distribution of selected strategies by ChatGPT. As for the reference responses, we observe that the seller tends to express their positive/negative sentiment as well as negotiate in a positive/negative manner. Differently, ChatGPT with standard and proactive prompting prefers to use hedge words or polite expressions (*e.g.,* please and gratitude), indicating that ChatGPT essentially plays a nice role in negotiation. ChatGPT with ProCoT prompting makes more decisions to use assertive words or trade in, compared with other distributions. This shows that ProCoT can enable ChatGPT to involve certain negotiation strategies.

| *Clarification Dialogues* |
| --- |

**Standard Prompting**: Given the document and the conversation history, generate the response.

**Proactive Prompting**: Given the document and the conversation history, answer the question or ask a clarifying question. The response should start with "The answer is" or "The clarifying question is".

**ProCoT Prompting**: Given the document and the conversation history, first identify whether the question is ambiguous or not. If it is ambiguous, ask a clarifying question. If it is not ambiguous, answer the question. The response should start with the ambiguity analysis of the question and then follow by "Therefore, the question is not ambiguous. The answer is" or "Therefore, the question is ambiguous. The clarifying question is".

**Sample**:

Document: "Angie went to the library with her mother. First she had to turn in the books she was returning at the return desk. They said hello to the man there. He took their books. Then they went into the adult reading room. Angie sat in a brown chair at the table. She made a drawing of her mother. Her mother found a large red book. Then they went to the Mystery section. Angie sat in a blue chair. She drew a picture of her brother. Her mother found the book. It was a green book. Finally it was time to go to the children's room. It was Story Hour. Miss Hudson was there to read to all the children. She read a book about friendship. After the story Angie sat in the red chair and began drawing. They were drawing pictures of friends. Angie drew a picture of her best friend Lilly. Miss Hudson hung the pictures on the wall. Then Angie and her mother picked out 8 books to read at home. They checked the books out and went home."

Conversation history: ["User": "What did she draw?", "System": "Her mother", "User": "What did her mother find?", "System": "The book"]

Question: "What color was it?"

**Demonstration (Standard)**: Do you mean the first book?

**Demonstration (Proactive)**: The clarifying question is "Do you mean the first book?"

**Demonstration (ProCoT)**: There are two books that book that Angie's mother found. It is uncertain which book is referred to. Therefore, the question is ambiguous. The clarifying question is "Do you mean the first book?"

Table 10: Examples of prompting LLMs for clarification dialogues.

| *Target-guided Dialogues* |
| --- |
| **Standard Prompting**: Given the target topic and the conversation history, generate the response. |
| **Proactive Prompting**: Given the target topic and the conversation history, predict the appropriate next topics that can bridge the current conversation topics to approach the target topics smoothly. Then based on the predicted next topics, generate the response. Please reply by completing the output template "The next topics are []. The response is". |
| **Proactive Prompting w/ CoT**: Given the target topic and the conversation history, consider the relationship between the current conversation topics and the target topics, and then predict the appropriate next topics that can bridge the current conversation topics to approach the target topics smoothly. Then based on the predicted next topics, generate the response. Please reply by completing the output template "The current topics are []. To bridge the current topics with the target topics, the next topics are []. Based on the predicted next topics, the response is". |
| **Sample**:
 Target topic: "Chicken"
 Conversation history: ["User": "I also remodel homes when I am not out bow hunting.", "System": "That's neat. When I was in high school I placed 6th in 100m dash!", "User": "That's awesome. Do you have a favorite season or time of year?"] |
| **Demonstration (Standard)**: I do not. But I do have a favorite meat since that is all I eat exclusively. |
| **Demonstration (Proactive)**: The next topics are ["eat", "meat"]. The response is "I do not. But I do have a favorite meat since that is all I eat exclusively." |
| **Demonstration (ProCoT)**: The current topics are ["season", "time", "year"]. To bridge the current topics with the target topics, the next topics are ["eat", "meat"]. Based on the predicted next topics, the response is "I do not. But I do have a favorite meat since that is all I eat exclusively." |

Table 11: Examples of prompting LLMs for target-guided dialogues.

| |
|---|
| *Non-collaborative Dialogues* |

| |
|---|
| **Standard Prompting**: Assume you are the seller. Given the item description, the target selling price, and the conversation history, generate the response. |

| |
|---|
| **Proactive Prompting**: Assume you are the seller. Given the item description, the target selling price, and the conversation history, in order to reach a better deal with the buyer, first select the most appropriate set of negotiation strategies and the most appropriate dialogue act to reach the bargain price. Based on the selected negotiation strategies and dialogue act, generate the response. The reply should be in the form "The most appropriate set of negotiation strategies is [] and the most appropriate dialogue act is []. Based on the selected negotiation strategies and dialogue act, the response is" |

| |
|---|
| **ProCoT Prompting**: Assume you are the seller. Given the item description, the target selling price, and the conversation history, in order to reach a better deal with the buyer, first analyse the current negotiation progress and consider an appropriate negotiation goal, then select the most appropriate set of negotiation strategies and the most appropriate dialogue act to reach the bargain price. Based on the selected negotiation strategies and dialogue act, generate a response. The reply should start with the analysis of the current negotiation progress and an appropriate goal, and then follow by "To reach this goal, the most appropriate set of negotiation strategies is [] and the most appropriate dialogue act is []. Based on the selected negotiation strategies and dialogue act, the response is" |

| |
|---|
| **Pre-defined Dialogue Acts**: [Answer a question, Proposing a counter price, Accept the offer, ...] |

| |
|---|
| **Pre-defined Negotiation Strategies**: [Communicate politely, Build rapport, Show dominance, ...] |

| |
|---|
| **Sample**: 
 Item description: "Phone charge two devices simultaneously on the go. This vehicle charger with an additional USB port delivers enough power to charge two devices at once. The push button activated led connector light means no more fumbling in the dark trying to connect your device. Auto detect IC technology automatically detects the device type and its specific charging needs for improved compatibility. And the built in indicator light illuminates red to let you know the charger is receiving power and the power socket is working properly. Verizon car charger with dual output micro USB and led light." Target selling price: 10. 
 Conversation history: ["Buyer": "Hi, not sure if the charger would work for my car. can you sell it to me for $5?", "Seller": "It will work, I have never seen a car without a cigarette lighter port.", "Buyer": "Still, can i buy it for $5? I'm on a tight budget."] |

| |
|---|
| **Demonstration (Standard)**: I think the lowest I would want to go is 8. |

| |
|---|
| **Demonstration (Proactive)**: The most appropriate set of negotiation strategies is ["Propose price", "Show dominance", 'Certainty words'] and the most appropriate dialogue act is ["Proposing a counter price"]. Based on the selected negotiation strategies and dialogue act, the response is "I think the lowest I would want to go is 8." |

| |
|---|
| **Demonstration (ProCoT)**: The buyer proposes a low price, which is unacceptable. The next step should assertively raise the bargain price. To reach this goal, the most appropriate set of negotiation strategies is ["Propose price", "Show dominance", 'Certainty words'] and the most appropriate dialogue act is ["Proposing a counter price"]. Based on the selected negotiation strategies and dialogue act, the response is "I think the lowest I would want to go is 8." |

Table 12: Examples of prompting LLMs for non-collaborative dialogues.