# OpenReview forum: "Prompting and Evaluating Large Language Models for Proactive Dialogues: Clarification, Target-guided, and Non-collaboration"
_EMNLP/2023/Conference — EMNLP 2023 Findings_

### Official Review · Reviewer_eUzu · 2023-07-25

**Soundness:** 3

**Excitement:**

4: Strong: This paper deepens the understanding of some phenomenon or lowers the barriers to an existing research direction.

**Missing References:**

No missing references.

**Paper Topic And Main Contributions:**

The paper introduces the first comprehensive analysis of Large Language Model-based conversational systems on three common aspects of proactive dialogues (clarification, target-guided, and non-collaborative dialogues).

Main contributions:
-The proactive chain-of-thought prompting scheme (ProCoT) is designed to endow the dialogue systems with the capability of planning and taking the initiative towards the conversational goal.
-Experiments with two LLM-based conversational systems, including ChatGPT and an open-sourced model (Vicuna) are carried out, the findings are analyzed.

**Questions For The Authors:**

No questions.

**Reasons To Accept:**

The main findings of the evaluation of LLM-based dialogue systems are explained:
- They barely ask clarification questions when encountering ambiguous queries;
- They tend to make aggressive topic transition.
- They fail to make strategic decisions.
It is shown how the proposed ProCoT scheme can help to overcome these limitations.



**Reasons To Reject:**

The paper should be accepted.

**Reproducibility:**

3: Could reproduce the results with some difficulty. The settings of parameters are underspecified or subjectively determined; the training/evaluation data are not widely available.

**Reviewer Confidence:**

3: Pretty sure, but there's a chance I missed something. Although I have a good feel for this area in general, I did not carefully check the paper's details, e.g., the math, experimental design, or novelty.

**Typos Grammar Style And Presentation Improvements:**

No typos.
However, the Appendix (pages 12 to 20, starting from line 962) includes the material that actually belongs to the main part of the paper (still, page limit is 8).

---

> ### Author Rebuttal · Authors · 2023-08-28
>
> We sincerely appreciate your acknowledgement of our work. We believe that this work can provide a comprehensive picture about how the current LLM-based dialogue systems can handle different types of proactive dialogue systems. It also discusses which aspects of the proactivity can be improved for further enhancing the LLM-based dialogue systems for proactive dialogues.
>
> Regarding the materials in Appendix, we greatly appreciate the suggestions. We will carefully reorganize the space. For example, we will move Appendix B to the main body by abbreviating the automatic evaluation results of non-collaborative dialogues as well as putting its full version into the Appendix. We strongly agree that this will make the analysis clearer.

---

### Official Review · Reviewer_e85V · 2023-08-05

**Soundness:** 2

**Excitement:**

3: Ambivalent: It has merits (e.g., it reports state-of-the-art results, the idea is nice), but there are key weaknesses (e.g., it describes incremental work), and it can significantly benefit from another round of revision. However, I won't object to accepting it if my co-reviewers champion it.

**Paper Topic And Main Contributions:**

The paper presents an analysis of LLMs (in particular ChatGPT and Vicuna) regarding their capabilities to generate proactive dialogue under different prompting schemes – standard, proactive and proactive chain of thought (ProCoT) and under zero-shot and one-shot conditions. ProCoT is defined by the authors to combine CoT with proactive strategies. The following three proactive dialogue capabilities are investigated: clarification dialogues in information seeking, proactively leading the conversation towards a designated target, and reaching a consensus with the user when user and system do not have the same conversational goal. The results are compared with results from fine-tuned models (baseline and SOTA) and the outcomes are discussed. The presented work makes use of a variety of methods and evaluation metrics. Which is an asset! The analyses result in a broad description of (mostly missing/weak) proactive dialogue capabilities of the two tested LLMs under the respective prompting schemes and 0/1-shot conditions. In other words, the work provides a picture of what is, but less so why.

**Reasons To Accept:**

Broad investigation of LLMs’ (ChatGPT, Vicuna) proactive dialogue capabilities under different prompting schemes and under zero-shot and one-shot conditions, as well as their comparison with state-of-the-art fine-tuned models.
The work offers a wealth of material and references related to proactive dialogue systems and their evaluation.


**Reasons To Reject:**

The paper is very dense and it requires quite some effort to reconstruct what has been done, e.g., it is not always clear what is in the tables, where the baseline and SOTA figures come from, how hard and easy targets in dialogue-level evaluation are defined, etc. (see Presentation Improvements). More streamlining and guidance would be required to make the paper ready for publishing in a conference. Moreover, the evaluation results show a rather mixed picture and it is hard to draw general conclusions.

**Reproducibility:**

2: Would be hard pressed to reproduce the results. The contribution depends on data that are simply not available outside the author's institution or consortium; not enough details are provided.

**Reviewer Confidence:**

4: Quite sure. I tried to check the important points carefully. It's unlikely, though conceivable, that I missed something that should affect my ratings.

**Typos Grammar Style And Presentation Improvements:**

A motivation should be given why you test in particular with ChatGPT and Vicuna, and not with other LLMs.

Line 173: PLM ?  LLM

Table 1: hard to reconstruct where the Baseline and SOTA figures come from (except for Abg-CoQA BLEU-1), clarify for all Baseline and SOTA figures which models were used and where exactly (reference plus where in the reference) the individual figures are published. Note there is also an inconsistency between Table 1 (BLEU-1) and Appendix B.1 which says BLEU-2 is used.

Line 296: provide details regarding the Help. score, such as how many annotators, what are the criteria used, what is the intercoder agreement.

Line 371 and Table 3: use of different terminology – next topic selection vs next topic prediction; this might be confusing for the reader, so stick to one term throughout the paper.

Line 388: COLOR is mentioned as one of the fine-tuned baselines for target-guided dialogues, however, it does not show in Table 3, briefly mention why.

Line 402: clarify what R-L stands for: ROUGE-L

Line 405-407 is not fully backed by the results reported in Table 3 where one-shot results for hits@1 are typically better than zero-shot results.

Line 410-425: also, here it seems the arguments are not fully reflected in the figures presented in Table 3.

Table 5: It needs to be explained why TopKG is the only finetuned system used for comparison with the LLMs?

All tables: make sure to provide all relevant information in the caption, e.g., for the measures what is good – high or low value; in Tables 3 and 4, mention which corpora are used; Table 4 and 5, briefly describe what is a hard/easy target.

Note, there are a few issues with subject verb congruency in the text.

---

> ### Author Rebuttal · Authors · 2023-08-28
>
> We appreciate the detailed and valuable comments. We will carefully address your concerns one by one as follows:
>
> 1. "The paper is very dense and it requires quite some effort to reconstruct what has been done"
>
> **Response**: We apologize that we have to condense all the content into 8 pages due to the space limitation. However, we believe that this issue can be easily and effectively adjusted by some reorganization. For example, as mentioned by the suggestion from Reviewer NW9t, we can abbreviate some less important evaluation results (e.g., the automatic evaluation results of non-collaborative dialogues) and put the full version into the Appendix, so that we could move the description of the adopted evaluation metrics to the main body. This will largely improve the presentation issue.
>
> 2. "the evaluation results show a rather mixed picture and it is hard to draw general conclusions"
>
> **Response**:  We did not claim to propose a “SOTA” method or aim to draw the same conclusions for all the problems. The target of this work is to (1) provide a comprehensive evaluation on how the current LLM-based dialogue systems perform in different proactive dialogue problems and (2) testify the effectiveness of different prompting schemes on triggering the proactivity of LLM-based dialogue systems. Thanks to the mixed picture of evaluation results, we could identify what kinds of proactive dialogue problems are difficult to be handled by LLM-based dialogue systems and which aspects of proactivity are lacking in current LLM-based dialogue systems. In order to make the take-aways clearer, we will add a road map to summarize the main findings for each prompting scheme and each proactive dialogue problem. This will be really beneficial for future studies on proactive dialogue systems in this era of LLMs.
>
> 3. “A motivation should be given why you test in particular with ChatGPT and Vicuna, and not with other LLMs.”
>
> **Response**:  We appreciate the suggestion. As for the motivation, we would like to evaluate both commercial black-box and open-source LLM-based dialogue systems. Before the EMNLP submission deadline, ChatGPT and Vicuna are the most representative LLMs that are trained for chatting in these two categories, respectively. We will make this motivation clearly discussed.
>
> 4. “PLM → LLM”
>
> **Response**:  Thanks so much for pointing out the typo.
>
> 5. "Table 1: hard to reconstruct where the Baseline and SOTA figures come from"
>
> **Response**:  We apologize for the confusion. We will add more clear descriptions in the table caption. Besides, thanks so much for pointing out the typo of BLEU-2 in Appendix B.
>
> 6. "Line 296: provide details regarding the Help. score"
>
> **Response**:  We conduct the similar human evaluation as other two tasks. Three annotators are asked to score whether the generated question is helpful for clarifying the existing ambiguity by 0 or 1. The Fleiss’ kappa of the annotations is 0.64, which indicates “substantial agreement”, and the final scores are calculated by average. We will make the human evaluation of the Help. score more clearly.
>
> 7. "Line 371 and Table 3: use of different terminology – next topic selection vs next topic prediction"
>
> **Response**:  Thanks so much for the suggestion of sticking to the same terminology of “next topic prediction”. We will revise all these terms to be “next topic prediction”.
>
> 8. "Line 388: COLOR is mentioned as one of the fine-tuned baselines for target-guided dialogues, however, it does not show in Table 3, briefly mention why."
>
> **Response**: Sorry for the confusion. In the current tables, we just present the reported experimental results from the original paper of each method. Since the model COLOR has not been implemented for the OTTers dataset, we did not present this result as this model was just released in June. In order to clarify the concern, we further implement the COLOR for the OTTers dataset with the open-source code. The results are as follows:
>
> |Method| Shot| Prompt| BLEU| METEOR| R-L| hits@1| hits@3|
> |---|---|---|---|---|---|---|---|
> |TopKG| -| -| 15.35| 13.41| 27.16| 7.78| 22.06|
> |COLOR|-|-|14.53| 13.70| 26.92| 8.02| 22.30|
>
> 9. "Line 402: clarify what R-L stands for: ROUGE-L"
>
> **Response**: We will clarify that R-L stands for Rouge-L.
>
> 10. "Line 405-407 is not fully backed by the results reported in Table 3 where one-shot results for hits@1 are typically better than zero-shot results."
>
> **Response**: Sorry for the confusion. In Line 405-407, we would like to present the fact that the one-shot ICL cannot contribute to a consistent performance improvement, while it may even cause some decreases in certain metrics. We will carefully modify the argument to make this point clearer.
>
> 11. "Line 410-425: also, here it seems the arguments are not fully reflected in the figures presented in Table 3."
>
> **Response**: We apologize for any confusion in this discussion. In order to make the statement clearer, Line 419-425 will be revised to be *“With one-shot demonstrations, ChatGPT with proactive prompts continues to underperform compared to standard prompts when it comes to accurately predicting suitable topics towards the target. However, it is worth noting that only ProCoT prompts consistently show an improvement in the performance of all LLMs for next topic prediction.”*
>
> 12. "Table 5: It needs to be explained why TopKG is the only finetuned system used for comparison with the LLMs?"
>
> **Response**: We would like to clarify that the goal is to compare the LLMs with the SOTA finetuned system. The latest SOTA, i.e., COLOR, was released in June, which was around the submission deadline of EMNLP. Therefore, in our main experiment, we still adopted TopKG as the SOTA finetuned system for experiments. In order to present an up-to-date result, we further conduct the same human evaluation on the generated dialogue from COLOR. The updated Table 5 is as follows:
>
> | |  | Easy | Target | Hard | Target|
> |---|---|---|---|---|---|
> |Method| Prompt| G-Coh.| Effect.| G-Coh.| Effect. |
> |TopKG| -| 1.42| 1.24| 1.21| 1.10|
> |COLOR|-| 1.38| 1.42 | 1.24 | 1.18 |
>
> 13. "All tables: make sure to provide all relevant information in the caption, e.g., for the measures what is good – high or low value; in Tables 3 and 4, mention which corpora are used; Table 4 and 5, briefly describe what is a hard/easy target."
>
> **Response**: Sorry for the confusion regarding this information. Due to the space limitation, we did not re-emphasize this information in the table caption. The used corpora in Tables 3 and 4 is OTTers and TGConv, respectively, as introduced in Line 390-391 and Line 427-428. Hard/easy target represents the hard-to-reach and the easy-to-reach target in the TGConv dataset, as introduced in Appendix A. We greatly appreciate the thoughtful and valuable suggestions regarding elaborating more details in the table captions. We will carefully add more detailed and relevant information to make the table clearer.
>
> 14. "there are a few issues with subject verb congruency in the text."
>
> **Response**: We will also carefully review the whole paper to address them.
>
> Overall, as the main concern is regarding the dense presentation due to the space limitation, we believe that we can effectively address this concern by taking into account your valuable suggestions. Again, thanks so much for the detailed reviews.

---

### Official Review · Reviewer_NW9t · 2023-08-07

**Soundness:** 4

**Excitement:**

3: Ambivalent: It has merits (e.g., it reports state-of-the-art results, the idea is nice), but there are key weaknesses (e.g., it describes incremental work), and it can significantly benefit from another round of revision. However, I won't object to accepting it if my co-reviewers champion it.

**Paper Topic And Main Contributions:**

In a typical dialogue system (task oriented or conversational), the system directly produces a response to the user’s previous contribution in a somewhat reactive manner. In contrast, a Proactive dialogue system has to plan ahead about its actions towards achieving certain goals (which may even be different from the user’s goals). Consequently, the final response of a proactive dialogue system will be generated conditioned on this chosen plan/action.

With recent advances in LLM-based dialogue systems such as ChatGPT & Vicuna, here authors propose to build a proactive dialogue system with prompt engineering techniques.

They conduct a systematic and comprehensive evaluation of proactive capabilities of dialogue systems built using prompted LLMs. Specifically they test for 3 proactive capabilities:
1. Classification Dialogues: identifying the need to resolve the ambiguity and generating the appropriate clarification question for resolving it.
2. Target-guided Dialogues: Identifying the correct relevant next topic on path to the target topic and generating a response which achieves the smooth topic transition.
3. Non-collaborative dialogues: Identifying the correct negotiation strategy, dialogue act and then generating a response based on that.

Authors explore 3 prompting strategies:
1. Standard: where an LLM is prompted to produce a response directly.
2. Proactive: Where an LLM is prompted to first select an intermediate action/plan and then generate a response conditioned on selected action/plan.
3. ProCoT: Similar to Proactive, but has additional chain of thought reasoning for selecting plan/action.
Authors try both Zero shot and One shot prompts of the above variants.

The main contribution of the paper is a systematic evaluation of these 3 proactive capabilities. Authors evaluate each task with automatic evaluation metrics as well as human judgments.

For Clarification dialogues, ProCOT prompting works reasonably well but there is still a large gap to SOTA performance.
For Target-guided dialogues, most LLM prompts are good at identifying the next topic. ProCOT strategy also produces smoother topic shift responses.
For Non-collaborative dialogues, LLMs struggle in identifying negotiation strategy & dialogue act and consequently the final response is also suboptimal.


**Questions For The Authors:**

A. Table 1, Please check the column headings. Why is BLEU-1 used for Abg-CoQA and ROUGE-2 for PACIFIC? Why are those not the same?

B. For all tables, can you report the number of examples evaluated? Are the main differences used in drawing conclusions statistically significant? Some corpus stats will be useful.

C. Line 419-422: This doesn’t seem true for Vicuna.

D. Line 445: “ProCOT enables smoother topic transition” this may hold true for automatic metric coherence, but it doesn’t seem to hold with human evals. Is the automatic coherence metric trustworthy then?


**Reasons To Accept:**

* Authors present a comprehensive evaluation of 3 capabilities of proactive dialogue systems achieved by various LLM prompting strategies.
* The evaluation protocol is sound.
* The paper is well written and easy to understand (Except Appendix B, see below).
* Readers of the paper can get a good idea of how effective LLM prompting can be towards building proactive dialogue systems and what challenges still remain unaddressed.


**Reasons To Reject:**

* The main weakness of the paper is limited novelty.
* Adding intermediate steps (like selecting a plan/action) or Chain-of-through reasoning are not novel ideas.
* The only novel thing here is application of these ideas for building proactive capabilities and a comprehensive evaluation.
* But the evaluation itself is producing mixed results. ProCoT strategy can help in some cases but not all and there is still a large gap to SOTA approaches.
* Given how sensitive LLM outputs are to prompts, there is a chance that this work may be of limited impact and interesting to a small audience.


**Reproducibility:**

4: Could mostly reproduce the results, but there may be some variation because of sample variance or minor variations in their interpretation of the protocol or method.

**Reviewer Confidence:**

4: Quite sure. I tried to check the important points carefully. It's unlikely, though conceivable, that I missed something that should affect my ratings.

**Typos Grammar Style And Presentation Improvements:**

Appendix B is crucial for understanding what evaluation metrics were used. These are basically the column headers for many tables in the paper. Without completely understanding those it’s hard to understand the conclusions drawn in the paper. Crucial material like this should not be pushed to the appendix.

In contrast, there are 6 different values for predicting negotiation strategies and 6 different values for predicting dialogue acts, which one should the reader pay attention to? This is too much information. Most of these numbers will be highly correlated with each other anyway. Is there anything surprising here that it needs to be part of the main paper? It’s not talked about in the main paper, so this can be pushed to the appendix.


Line 173, What is a PLM?

---

> ### Author Rebuttal · Authors · 2023-08-28
>
> We appreciate the detailed and valuable comments. We will carefully address your questions one by one as follows:
>
> - Question A: "Table 1, Please check the column headings. Why is BLEU-1 used for Abg-CoQA and ROUGE-2 for PACIFIC? Why are those not the same?"
>
> **Response**: Sorry for the confusion. Since the previous works on Abg-CoQA and PACIFIC adopt different automatic evaluation metrics to present the result, we just follow the one that is commonly adopted in previous studies for each dataset. We will make it clearer.
>
> - Question B: "For all tables, can you report the number of examples evaluated? Are the main differences used in drawing conclusions statistically significant? Some corpus stats will be useful."
>
> **Response**: We apologize for the confusion about the dataset statistics. The number of dialogues evaluated for each dataset is presented in Appendix A (Table 8). Furthermore, we will add the significance test for better presenting the differences.
>
> - Question C: "Line 419-422: This doesn’t seem true for Vicuna."
>
> **Response**: Sorry for the confusion about the statement. We will further point out that this statement is for ChatGPT. Line 419-425 will be revised to be *“With one-shot demonstrations, ChatGPT with proactive prompts continues to underperform compared to standard prompts when it comes to accurately predicting suitable topics towards the target. However, it is worth noting that only ProCoT prompts consistently show an improvement in the performance of all LLMs for next topic prediction.”*
>
> - Question D: "Line 445: “ProCOT enables smoother topic transition” this may hold true for automatic metric coherence, but it doesn’t seem to hold with human evals. Is the automatic coherence metric trustworthy then?"
>
> **Response**: As shown in Table 5, the human evaluation results also demonstrate that “ProCOT enables smoother topic transition”, since the G-Coh scores of ProCoT are much higher than those of Standard.
>
> As for the mentioned concerns, please find our responses as follows:
>
> 1. Regarding the concern on the novelty of this work, as presented in the contributions, the main contribution of this work is to conduct a timely, innovative, and comprehensive evaluation on the proactivity of the LLM-based dialogue systems. Although the idea of Chain-of-Thought is not new, it is the first study to adapt this idea into the dialogue problem.
>
>
> 2. Regarding the concern on the mixed results in the experiments, we did not claim to propose a “SOTA” method for all the problems. The target is to (1) evaluate how the current LLM-based dialogue systems perform in different proactive dialogue problems and (2) testify the effectiveness of different prompting schemes on triggering the proactivity of LLM-based dialogue systems. Thanks to the mixed results, we could identify what kinds of proactive dialogue problems are difficult to be handled by LLM-based dialogue systems and which aspects of proactivity are lacking in current LLM-based dialogue systems.
>
>
> 3. Regarding the concern on the sensitivity of prompts, we conducted a sensitivity analysis of prompts in the Appendix D. In order to facilitate the reproducibility, we also present all the finalized prompts in the Appendix D and will release the prompt to the public. As the first study to evaluate the proactivity of LLM-based dialogue systems, we believe that it can serve as an easy-to-apply baseline for future studies.
>
>
> 4. Regarding the presentation, we sincerely appreciate your suggestions. In the revision, we will move Appendix B to the main body by abbreviating the automatic evaluation results of non-collaborative dialogues as well as putting its full version into the Appendix. We strongly agree that this will make the analysis clearer.
>
>
> 5. “PLM → LLM”: Thanks so much for pointing out the typo.

---

### Meta-Review · Area_Chair_99iq · 2023-10-06

**Recommendation:** 3

**Metareview:**

The paper presents an exploration of proactive dialogue systems using prompted LLMs, with a systematic evaluation of their capabilities. While the paper makes interesting contributions, there is a lack of reproducibility of the results since they rely on very specific prompting techniques.

---

### Decision · Program_Chairs · 2023-10-07

**Decision:**

Accept-Findings

**Comment:**

The paper presents an exploration of proactive dialogue systems using prompted LLMs, with a systematic evaluation of their capabilities. While the paper makes interesting contributions, there is a lack of reproducibility of the results since they rely on very specific prompting techniques.